# Transport, Mobility and the Wellbeing of Older Adults: An Exploration of Private Chauffeuring and Companionship Services in Malaysia

**DOI:** 10.3390/ijerph20032720

**Published:** 2023-02-03

**Authors:** Abdul Rais Abdul Latiff, Saidatulakmal Mohd

**Affiliations:** School of Social Sciences & USM Initiatives on Ageing (USIA) Research Cluster, Universiti Sains Malaysia, Gelugor 11800, Pulau Pinang, Malaysia

**Keywords:** private chauffeuring, companionship, transport, mobility, older persons, wellbeing

## Abstract

As physical abilities and health decline with age, older adults tend to lose their driving abilities, which affects their mobility. As mobility is important to older adults’ wellbeing, there is a need to explore alternative modes of transportation to increase their ability to actively participate in society. Hence, this paper aims to understand the characteristics of private chauffeuring and companionship services for older adults, and to assess their possible effects on older adults’ wellbeing. We gathered the views of transport operators, government agencies, and city councils that offer private chauffeuring and companionship services for older adults. We frame the model of private chauffeuring and companionship services as alternative mobility for older adults and outline a conceptual framework for its possible effects on their wellbeing. The underlying mobility characteristics were availability, accessibility, safety, and affordability—all of which influence wellbeing. The study found that the private chauffeuring and companionship model for older adults includes an additional model of government-to-consumer services in addition to the existing peer-to-peer and business-to-consumer services. While the services are available, the services provided are not standardized, with different operators offering different services and prices, and limiting certain geographical areas. Transport operators perceived that the services they offer promote older adults’ physical and mental health, improve their social participation in the community, and empower them in making their travel decisions. The findings of the paper provide insights for policy makers for future planning of alternative transportation for older adults to enhance their mobility.

## 1. Introduction

The population of adults aged 60 and above is expected to be 2.1 billion by 2050—double the number recorded in 2020 of 1.4 billion [1]. As part of an effort to support older adults’ continued involvement in families and communities, [2] introduced an ‘active ageing’ concept to optimize opportunities for health, participation, and security to enhance the quality of life of older adults. One of the enablers of active ageing is mobility [3]. Defined as a person’s ability to move from one place to another [4], mobility is considered an important element of active ageing for good quality of life [5,6] that improves wellbeing [7].

Driving remains the main choice of transportation for older adults [8,9,10] to maintain autonomy and self-esteem [11]. However, driving in old age poses great challenges, especially safety and mobility concerns, due to the physical and cognitive changes associated with ageing [12,13,14]. Older adult drivers are connected to higher crash fatality rates attributed to increased frailty and injury susceptibility [15]. As a result, older adults often depend on family and friends [16] or public transportation to satisfy their mobility needs. Although many children of older adults tend to regard meeting their parents’ transportation needs as a moral responsibility, they want to ensure that more flexible mobility options are also available [17]. Hence, the provision of a transport system that fulfils older adults’ mobility needs, preferences, and demands is important [18].

Despite efforts to improve the quality, accessibility, and affordability of public transportation to meet the needs of older adults [19], the current transport system is still insufficient to support the increasing numbers of the ageing population [20] and unresponsive to older adults’ specific needs [21]. The infrastructure and facilities of public transport systems mainly cater to the needs of the young and middle-aged groups and ignore the travel needs of older adults [22]. Travel and mobility are not experienced in the same way by older adults as they are by young adults, as older adults have different travel characteristics regarding their travel purposes, modes, and distances [23]. Compared to young and middle-aged groups, older adults tend to make fewer trips and travel shorter distances with limited activities [24]. Public transportation may not cater to such needs, and this limits the accessibility of goods and services and continuous participation and involvement in the community for older adults, which could also affect their wellbeing.

Transportation mobility is important for life satisfaction as it promotes independence, good health, quality of life, wellbeing, and social integration for older adults [6,25]. The accessibility and availability of elderly-friendly transport is a prerequisite to older adults being able to live independently in a community [26] and avoid the kinds of social isolation, depression, and loneliness that often impact the wellbeing of older adults [27]. A previous study [28] conceptualized three ways in which mobility influences wellbeing: first, mobility allows older adults access to destinations and social connections that improve their health and wellbeing; second, it improves physical health through physical activities; third, it enhances the feeling of wellbeing. An inability to use public transport or no longer being able to drive denies older adults’ participation in social activities. As a result, some older adults experience transport disadvantages and social exclusion [29]. According to [30], older adults do not utilize alternative transportation for five main reasons: acceptability, affordability, adaptability, and accessibility. Therefore, transport and mobility need to take these deficiencies into account and accommodate safety features. A previous [31] identified five components of transport provision for an ageing society, namely, housing and transport, travel availability, transport accessibility, transport affordability, and transport safety. Transportation that fulfils these elements ensures that older adults improve their quality of life through social inclusion and independence in performing daily activities, which enhances their wellbeing and health, promotes life satisfaction through the comfort and convenience of aged-friendly transport, and encourages equity by not leaving older adults behind.

One alternative transportation that falls within the transport framework is private chauffeuring and companionship. Private chauffeuring (also called escorting, service passenger, and caregiving travel) refers to vehicle travel specifically for transporting non-drivers [32]. Chauffeuring can include commercial transport, such as taxi services, as well as household chauffeuring: incremental unpaid motor vehicle travel specifically made to transport family members or friends who are independent non-drivers (people capable of independent travel if suitable mobility options are available) [32]. Private chauffeuring, similar to ride-sourcing, ride-hailing, on-demand rides, for-hire vehicles (FHVs), and transportation network companies (TNCs), has emerged as a mobility option in many cities worldwide, offering personal door-to-door services to its passengers [33,34] and addressing some of the transport-related challenges faced by society [35]. Passengers schedule and book their trips in advance via phone, websites, or smartphone apps, street-hail by raising a hand on the street or standing at a taxi stand or specified loading zone, or e-hail by dispatching a driver on-demand using a smartphone application [33]. Based on bookings received for pick-up and drop-off locations, TNCs match passengers who require a ride with self-employed drivers who offer transportation services in their privately owned cars [36].

Older adults, constrained by their ability to drive, benefit from ride-sourcing services for transportation accessibility [37,38] that enhance their mobility freedom without depending so much on friends or family members for their travel needs [39]. Older adults with medical conditions such as visual and hearing impairments, mobility impairments, and chronic health conditions prefer door-to-door or door-through-door services (personal and hands-on assistance) and services that include accompanying them to appointments or when running errands [40]. Ride-sourcing is an excellent option as older adults require transportation routes aligned with healthcare facilities and traveling with public transportation often requires multiple stops and walking some distance from the bus stop or transportation hub, which can be physically challenging [41].

With the change in technology and the widespread use of smartphones, ride-sourcing services, through apps, have gained importance as transportation modes for older adults. As older adults have specific needs related to their frailty and vulnerability, drivers on demand such as private chauffeurs and companions could be a good option to facilitate the mobility needs of older adults. Two research questions to be addressed by this paper are (i) what are the characteristics of private chauffeuring and companionship services that best suit the needs of older adults, and (ii) what are the effects of mobility that are motivated by private chauffeuring and companionship services on older adult’s wellbeing. As such, this paper aims to understand the characteristics of private chauffeuring and companionship services for older adults and to assess its possible effects on their wellbeing.

## 2. Review of Empirical Studies

In many countries, older people’s travel behavior is mainly characterized by car dependency as primary transport either as driver or passenger [42,43]. Nevertheless, as physical abilities and health decline with age, older adults tend to lose their driving abilities. A previous study [42] found that car driving dramatically declined for those aged 85 years and more, and was replaced with being a car passenger, walking, and transit modes. In urban areas, the concept of car sharing in the form of private chauffeuring, ride sourcing, ride sharing, and on-demand rides has started to gain importance [44]. The number of car shares is estimated to grow to 36 million by 2025 from 7 million in 2015 [45], as cited in Hjorteset and Böcker [46]. One business model of car sharing includes peer-to-peer (P2P) services, sharing privately owned vehicles for a particular trip [47,48,49]. P2P includes chauffeur or taxi services on demand, matching privately owned vehicles to users on demand through smart phone technology or social networks [44]. Another model is business-to-consumer (B2C) services where firms or organizations provide the on-demand services to consumers [48,49]. Two other models are business to-government (B2G) and business-to-business (B2B) models [49]. B2G is a car-sharing service offering transportation services to a public agency, while B2B is a car-sharing service offered to employees to complete work-related trips.

One characteristic of car-sharing services is the location they serve. Car-sharing services are becoming popular in urban areas in which car sharing is regarded as a transport alternative [50]. There are more car-sharing networks within areas near the city than in suburban areas [51], allowing users to avoid congestion, parking, and pollution problems in addition to accomplishing several transportation goals [52]. While suburban and rural areas are seen as potential markets for P2P car sharing, setting up of P2P in rural areas can be more expensive and inconvenient than in urban areas, as rural markets are still underserved due to limited supply [53]. Recent evidence shows that demand for car-sharing services are also prominent in suburban and rural areas [54] as tools to meet the mobility needs of the residence [55].

The growth in car-sharing services has benefited significantly from the development of digital technologies [51,56,57] and has evolved into a more collaborative and self-service approach [58]. Reservation and payment of services could be performed ahead of time via digital platforms, social media, or social network websites. Ease of reservation and payment is perceived as appealing to car-sharing services’ users [59]. The social benefits associated with car-sharing services include greater accessibility of services to a larger group of customers from geographically isolated areas [53]. Smart technology also allows for travel monitoring as an additional feature to guarantee the safety of passengers and drivers. As found by Shaheen, Martin, and Bansal [53], one of the barriers to car-sharing services as faced by operators is building trust with customers, and technology could overcome this barrier.

The safety features of car-sharing services include the safety of users and the safety of the vehicle itself. Car-sharing service providers should take appropriate and effective measures in ensuring that their vehicles undergo strict examinations, are equipped with the necessary insurance, and have enhanced supervision [60]. Concerns over safety of car-sharing services also include fear of sharing with strangers, personal safety, and driving characteristics [61]. A B2C car-sharing model such as Uber has a diverse set of safety features for the vehicle, drivers, and customers [62]. Insurance is an important feature, to protect both users and vehicle owners against accidents, damage, theft, and robbery. Some car-sharing operators extend insurance to their customers, while some require that customers have appropriate insurance policies [53]. Customers traveling with B2C car-sharing services such as Uber X are protected by insurance [62].

Another characteristic of car-sharing services is the lack of standardization in terms of prices among P2P car-sharing operators. Car-sharing operators believe that hosts can set their own prices depending on the market [53]. Different prices are observable as the car-sharing services provided may vary depending on customers’ needs, such as distance, waiting time, availability of companionship, and additional safety features. By adopting the MATSim framework to assess the interaction of car-sharing competitors, Balac et al. [63] found that different price structures attract different market segments and different strategies, and price levels can affect the service usage and profitability. Nevertheless, costs should be set at a reasonable price, as cheaper costs ensure door-to-door services are more attractive and affordable [57,59].

Car-sharing services also contribute to social factors, enhancing and enabling mobility among those who are unable to drive themselves [57]. In the case of older adults, car-sharing services allow older adults to gain their independence and flexibility to make trips and increase their mobility [64]. The demand-responsive model, which offers door-to-door travel services, is an alternative mobility service provided either by private or public sectors for older adults to meet their specific demands [65], and thus enhance and improve their wellbeing through mobility. Not-for-profit providers offer various types of assistance, such as helping older adults in and out of the vehicle, offered a steadying arm, and waiting for older adults to complete their appointments or errands, thus empowering older adults, enabling them to be independent and engaged in the community, enhancing their health, and improving their quality of life [12]. A similar characteristic is observed among for-profit car providers who provide reliable and affordable transportation to improve older adults’ lives [12].

On-demand ride-hailing services have increased older adult’s travel trips and have potential to serve as a complementary form of public transportation for some older adults [39]. Integration of car-sharing services into the public transit system enables the two to work together [55] in providing good mobility services to older adults to enhance wellbeing. Importantly, car-sharing services should fulfil older people’s mobility attributes, such as the psychological benefits of movement, exercise benefits that enhance physical and mental health, involvement in the local community to ensure social connectedness, and the potential to travel [66,67].

## 3. Methodology

### 3.1. Participants

Participants of this study included representatives from private transport providers (PTs), land transportation agencies (TAs), and city councils (CCs) of major cities in Klang Valley, Malaysia. In accordance with the qualitative research approach, the study employed non-probability purposive sampling. Researcher made a list of PTs, TAs, and CCs in Klang Valley. There are a total of 10,720 PTs (6846 taxis/rented cars and 3874 bus services), 2 TAs (Land public transport agency and Ministry of Transport), and 11 CCs in Klang Valley. From the list, the study invited both TAs and all CCs to FGDs. A total of 6 FGD invitations were sent to PTs, which were randomly picked from the list representing taxis/rented cars and bus services. The total number of invitations sent was 50 and 42% responded. A total of five PTs, five TAs, and 11 CCs attended the FGDs.

### 3.2. Ethical Consideration

Ethical clearance for the study was received from the Universiti Sains Malaysia’ ethnical committee. Participants’ names were not documented during the data collection process and only the demographic data were documented in the informed consent form.

### 3.3. Data Collection

Three series of focus group discussions (FGDs) were conducted with PTs on 12 May 2022 and 26 September 2022; with TAs on 12 May 2022; and with CCs on 12 May 2022 and 11 August 2022. The 12 May session with PTs was conducted face-to-face, while the 26 September session was conducted online. The sessions with the TAs and CCs were conducted online for the 12 May session and face-to-face for the 11 August session. Only relevant quotes and points of discussion are included in this paper.

The protocol of the FGD session, whether conducted online or face-to-face, was as follows:

Step 1—The facilitator welcomed participants to the session. In accordance with ethic principles, participants were first requested to consent to their participation for the session. A link to an online informed consent form was shared with the participants, in which the facilitator read the ethical concerns and data privacy and confidentiality issues to the participants. A rapporteur assisted with any accessibility issue in accessing the consent form. The facilitator then moved to the next step once all participants had completed the consent form.

Step 2—The facilitator explained the purpose of the sessions and elaborated on how the session was to be conducted. The facilitator informed participants that the session would be recorded and that the discussion notes would be taken down by a rapporteur, for which the notes would be typed directly in Google Slide. The Google Slide was projected on the screen so that participants could view what was written down by the rapporteur and could suggest amendments to the notes, if any. Participants were also provided the flexibility to type their responses directly into the Google Slide, for which access to the slide was provided to all participants. The agenda of the session and the approximate minutes for the discussion were shared with participants.

Step 3—The discussions during the FGDs were divided into two major discussions: (i) views on private chauffeuring and companionship to fulfil older persons’ mobility needs, and (ii) challenges faced in meeting older persons’ mobility needs. The key questions for each topic are shown in Table 1 below.

Step 4—Once the discussion was over, the facilitator thanked all participants. Each participant was provided a small token of appreciation as an acknowledgement of their valuable time and input.

### 3.4. Data Analysis

As a qualitative study, the data analysis involved content analysis of the notes and transcriptions from the FGDs. Content analysis of the input provided by the study participants was conducted to describe, interpret, and deduce the type of private chauffeuring and companionship services provided. In this paper, private chauffeuring and companionship refer to services in which a driver provides transport in a private vehicle, or someone rides with the older adult as a companion from their origin to their destination. The origin could be from a transportation hub such as a taxi stand or one’s residence. Reservation of the ride could be made by telephone, hailing from a designated location, standing at a transportation hub or taxi stand, or using a mobile application. Services could be provided either by a private transport operator, public transport operator, or the city council. The lead researcher finalized the notes in the Google Slide and played back the audio recordings to clarify certain points that were not clearly written on the Google Slide. All text was then uploaded into NVivo for analysis. The themes for the study’s first objectives were deductively identified based on [31] transport provision for older adults: (i) availability, (ii) accessibility, (iii) safety, and (iv) affordability, which were also the topics discussed during the FGDs.

The themes for wellbeing were also deductively identified following Ingersoll-Dayton et al. [68], which suited our purpose for two main reasons. First, this previous study [68] examined older adults’ psychological wellbeing by developing psychological measures that are relevant to indigenous local culture. Qualitative measures were used to identify the constructs of wellbeing prior to measuring wellbeing quantitatively. The five constructs formed were harmony, interdependence, respect, acceptance, and enjoyment. Harmony encompasses elements on the relationship between older adults with family members and the community; interdependence measures older adults’ reliance on family members; community respect measures the behavior of younger family members towards older adults; acceptance includes measures of older adults’ perception of themselves in handling certain situations; and enjoyment refers to older adults’ experience of happy and pleasant times. The wellbeing constructs incorporated measures of Asian values that were different from those of Western culture. Second, although the previous study involved older adults, which was different from the current study’s participants, the wellbeing constructs represented the measures of psychological wellbeing of older persons that could be adopted to enhance one’s understanding of cross-cultural variation in older adults’ wellbeing. We measured companionship as a proxy for interdependence, empowerment as a proxy for respect, mental health as a proxy for acceptance, physical health as a proxy for harmony, and social engagement as a proxy for enjoyment. Companionship has the characteristics of dependency of older adults on the companion for mobility, as interdependence measures older adults’ reliance on family members and community. Empowerment refers to the ability of older adults to make decisions through active participation in social activities and to be perceived as independent, as respect measures the behavior of younger family members towards older adults. Mental health refers to older adults’ ability to sustain positive emotion through various social engagements, as acceptance includes measures of older adults’ perception of themselves in handling certain situations. Physical health relates to older adults’ engagement in activities with family members and the community through their mobility, as harmony encompasses elements of the relationship between older adults and family members and the community. Social engagement refers to older adults’ positive experience from being mobile, as enjoyment refers to older adults’ experience of happy and pleasant times.

The text was then classified into their appropriate themes as researchers analyzed the text. Three rounds of text analysis were conducted. First, the corresponding author analyzed the text and categorized it into suitable themes. Then, a discussion was conducted with the lead researcher to conform the text and its appropriate themes. Finally, the findings were presented to a group of experts from the Consortium on Mobility and Transportation in an Ageing Society (CoMTAS) to reaffirm the researchers’ analysis. Following Malterud, Siersma, and Guassora [69], data sufficiency was attained when sufficient information was gained by considering the study aim, sample of participants, quality of the dialogue and text taken during FGDs, and the data analysis strategy. As more text from the various FGDs was analyzed, the contents for the chosen themes began to repeat from the data analysis, and we understood that we have reached saturation stage and met the study’s aim.

### 3.5. Trustworthiness

We adopted the trustworthiness framework by Lincoln and Guba [70] in assessing and preparing reliable and valid qualitative research, which encompassed credibility, dependability, conformability, and transferability. For credibility, we adopted data triangulation techniques, in which responses from our FGDs were analyzed and compared with the responses received in face-to-face sessions and online sessions. Researchers sat down to assess the data and information to provide the correct interpretation of older adults’ mobility, particularly on the concept of private chauffeuring and companionship services. A few discussion sessions among researchers and experts from the CoMTAS were conducted physically and online to discuss the process of data collection prior to the FGDs, in addition to data analysis and interpretation. A few changes were made during the discussions to better improve data collection and interpretation. This was also part of the data conformability process where findings were adjusted to reflect the nature of the participants under study. During this stage, contents for specific themes were identified to reflect the underlying mobility characteristics and private chauffeuring and companionship services for older people. Finally, the data were then transferred to NVivo for analysis.

## 4. Findings

### 4.1. Private Chauffeuring and Companionship Services for Older Adults’ Mobility

#### 4.1.1. Availability

Private chauffeuring and companionship services for older adults are already available in the market, provided by private transport operators following the P2P and B2G business models. As city councils also provide such services, another government-to-consumer (G2C) model emerges where a government agency shares the responsibility with private operators in providing on-demand door-to-door services to older adults in their areas. The provision of services by private operators and government stems from understanding the special needs of older adults, which could not be fulfilled by the current public transportation. A representative from the government transport agency believes that the availability of private chauffeuring and companionship services is due to catering such services to people with disabilities. One representative mentioned that “by referring to the facilities required by the disabled that have been provided, older adults can benefit from the available ramp and wheelchair … ensure that door-to-door transportation services are available to older adults … especially to meet the need to travel from a residential area to a transportation hub” (participant 9, government transport agency). This view was shared by a city council representative. The door-to-door services offered to families with disabled family members are also available to accommodate the transportation needs of older adults. The available transport is ready to accommodate older adults who use wheelchairs. The service provided is explained by one of the city council representatives: “We have three vans based on request. If older adults have a doctor’s appointment and require transportation, we can pick them up at their house and take them to the hospital. But, they must inform us seven days in advance. The demand for such service is high—an average of 30 trips per month. … we also extend this pick-up and drop-off service to chauffer older adults to whichever destination they requested. … our van can accommodate wheelchairs for older adults who use wheelchairs” (participant 14, city council). Another city council considers the type of door-to-door transportation that is available to older adults, especially those who are experiencing mobility challenges. However, the reservation of their services is more simple. As explained by the city council, “old-age-friendly transport reservation services can be reserved through phone calls and not only through online reservations or waiting for public transportation, such as a bus or taxi, at the roadside” (participant 18, city council).

A more specific service that includes companionship is offered by private transport operators. One private transport operator provides private chauffeur services, including companionship with the help of freelancer drivers who use their own private vehicles for the service. They explained, “We provide chauffeuring and companionship services to older adults, women, and special needs children who experience transportation difficulties. … among the needs of older adults are wheelchair access, taking food or medicine at certain hours during the duration of services, and hand-holding for hospital services and toilet visits. All companionship services are agreed upon with the companion (a freelancer who acts as chauffeur and personal carer) prior to the confirmation of services” (participant 5, private transport operator). This findings were aligned with previous studies [12,65] that door-to-door transport operators extend specific services to older adults to ease their traveling needs The roles of the companions were clearly explained during the reservation, in which the transport operator would find an appropriate freelancer to provide such services. In the case of specific tasks, those with certain certifications were sought to ensure that the older persons would receive the required services.

#### 4.1.2. Accessibility

The availability of such services should be complemented by ease of accessibility for older adults. Despite the development of technologies [51,56,57] that have increased the growth in on-demand rides, some older adults prefer to use traditional methods of transport reservation through call centres, telephone calls, and short messages; these methods should be allowed for greater accessibility of services. Private taxis still operate quite traditionally, as customers can queue at the taxi stand to get the taxis they require. As explained by one taxi provider, “We used a queuing system at our taxi stand. Taxis are normally ready and available to transport passengers. If there are older adults queuing, priority will be given to them for the services. … we also used our call centre system where older adults can call our call centre to make a reservation for private door-to-door service” (participant 2, private transport operator).

Some transport operators already have regular customers who require their services at specified times, either for regular hospital visits or other purposes. For these customers, a reservation is often made via phone calls. One private transport operator mentioned that “We also have regular customers for such service. They will call us to inform us when they need the service and we will go and pick them up at home. We will then bring them to their destinations and wait for them to finish their errands before taking them back home” (participant 2, private transport operator). In some instances, where older adults are unable to make the reservations themselves, their children would make the reservations on their behalf. As explained by one private transport operator, “services are reserved through phone calls either by older adults themselves or their children. Once we have received a service request, we contact our freelancers who are available and compatible to execute such services. … We normally take requests at least 24 h in advance, particularly if there is demand for companionship in addition to just chauffeuring” (participant 4, private transport operator).

Although private transport operators are ready to provide private chauffeuring and companionship services to older adults, their services are sometimes limited to mobile older persons. One transport operator emphasised, “We provide services to older adults, but we cannot accommodate special needs such as wheelchairs … our vehicles are not fit for wheelchairs” (participant 1, private transport operator). In many instances, even if a request for transportation does not include companionship services, the services are provided out of courtesy to the older adults by the drivers. This was emphasised by one transport operator who mentioned that “Although special services are never included in the taxi fare, our drivers normally extend their services to the older adults, such as opening the doors, helping older adults to get in and out of the taxi, accompanying them to the hospital seating area, ensuring that they meet the intended person at the correct place, and waiting for the older adults to complete their appointment to chauffeur them back home” (participant 1, private transport operator).

#### 4.1.3. Safety

Although safety is an important feature of private chauffeuring services, some transport operators admit that there are no specific safety features to cater to the needs of older persons. One private transport operator mentioned that “We do not have any specific safety feature for older persons. The safety of traveling with our vehicles is the same either for older persons or other individuals” (participant 2, private transport operator). However, because private vehicles are often used to chauffeur older adults, operators must ensure that their vehicles are in good condition, have valid insurance, and follow the rules and regulations set by authorities for private vehicles, as emphasised by Min and Xing-Fu [60]. As part of the safety features, one private transport operator mentioned that “All our taxis must go through road transport inspection to ensure the safety of the vehicle. Only upon clearance of the inspection are we allowed to operate” (participant 1, private transport operator).

Nevertheless, some private operators adopt specific safety features to ensure the safety of older adults. One private transport operator explained their safety features as follows: “For safety in the vehicle, we ensure that the number of passengers follows the limit allowed per vehicle, seatbelts are worn by passengers, and COVID-19 safety precautions are followed by the chauffeur and passengers. Chauffers also inform family members when the older adults have arrived at their destinations. … We also recommend that our passengers take extra insurance of RM3 using eWallet” (participant 4, private transport operator). A similar finding with Shaheen, Martin and Bansal, [53] indicate that safety features include insurance for users. We also share the live location of the travel from pick-up to destination with family members so that they can track the location of the older adult.

#### 4.1.4. Affordability

Another characteristic of private chauffeuring services for older adults is the affordability of services. Since this service is privately provided, there is no standard fixed price set by the authorities and each operator decides on their own prices, similar to the findings of Shaheen, Martin, and Bansal [53]. As mentioned by one transport operator, “There is no special price for older adults using our services. Our price is standard—the same for all users” (participant 3, private transport operator). This situation was confirmed by another transport operator who mentioned that there was no fee reduction for services offered to older adults. He mentioned that “We do not have any special pass for older adults. Even for regular customers, there are no seasonal passes. As we are private operators, we take cash from our passengers, including older adults” (participant 1, private transport operator). Understanding that the services are private, intended for one person with a specific destination and task, the transportation charges are slightly higher than those of other public transportation. Although charges for private cars are monitored by the authority, there are no specific standards. This was mentioned by one transport operator: “Our charges are normally higher than other public transportation. We are unable to reduce the meter charges as the charges are set by the authorities. While the charges are monitored every month, they are fixed based on specific standards” (participant 2, private transport operator).

Price also varies based on distance and the kinds of services offered. As such, price is sometimes determined through a market survey to ensure that the prices quoted are competitive. Yet, the prices are not regulated by authorities and are set based on agreement with the passengers. This was iterated by one transport operator, who mentioned that “Price is determined by a market survey done to determine the prices to be charged for the types of services provided. The price is not regulated by the government. Reservation is confirmed when older adults or family members agree to the price quoted for the services required” (participant 5, private transport operator). Understanding the high costs of such services, some private transport operators do make exceptions for older adults from the low-income group. This was explained by one transport operator who mentioned that “We raised public funds to pay for older adults’ travel costs. … We also link passengers with certain agencies to cover the cost. … for older adults who are unable to pay the quoted prices, we help in finding them sponsors to bear the cost. … In certain instances, when all attempts fail, we will find another service provider for the older adults” (participant 5, private transport operator).

Depending on the situation, some older adults from low-income groups were not charged for the services they sought. This was confirmed by one transport operator who mentioned that “Sometimes, we do not charge the older adults. Especially those from a low-income category. We just allow them to use our services for free” (participant 1, private transport operator). In some instances, free services were offered to older adults who required private transportation. Nevertheless, the distance for the services offered is limited to certain areas. One city council representative mentioned that “Our service for older adults is free. … but our destination is limited to only within Shah Alam [the council area]” (participant 14, city council).

### 4.2. Private Chauffeuring and the Wellbeing of Older Adults

Understanding the travel needs of older adults is one aspect of taking care of their wellbeing that fullfills older adults’ mobility attributes [66,67]. This is emphasised by a government transport agency representative, who mentioned that “We prioritise older adults in our transportation policy. There have been a lot of engagement sessions with older adults to understand their transportation and mobility requirements. … older adults prefer to travel on their own and do not like assistance” (participant 9, government transport agency). In fact, many are aware that older adults need to continue to participate in their community and that public transportation may not meet their travel needs or safeguard and enhance their wellbeing. As emphasised by one city council, “Older adults prioritize the health, recreation, and daily needs that they can fulfil in their residential area. Public transportation is not preferred by older persons because they have their own preferences. … Older adults prioritize safety, time-saving, direct location, and non-stop transportation to maximise their travel and outdoor activities” (participant 11, city council).

Private chauffeuring is seen as a required service for older adults to ensure their safety, as older adults often travel alone without companions to retain their independence in performing daily tasks. Therefore, having a companion travel with the adult will ensure that their wellbeing is taken care of. This was emphasised by one city council member, who mentioned that “There is a need for special services, such as car services, to accommodate older adults traveling because they are often alone, and this is better for their mobility needs” (participant 15, city council). Private chauffeuring is not meant to replace the traditional role of children in caring for their parents, but to provide a way of empowering older adults to continue participating in society. Private chauffeuring and companionship can not only take older adults to their destinations but also assist in their activities, such as having a companion at hospital appointments or grocery shopping. One transport operator agrees: “Our role is not to replace older adults’ children or take over the children’s responsibilities towards their parents; our role is to empower the older adults by catering to their mobility needs” (participant 5, private transport operator).

One private transport operator described private chauffeuring services as services to ensure that the wellbeing of older adults is enhanced in many dimensions, which include health, inclusivity, and equity. They stated, “Our motto is to bring older adults outside of their home so that they are healthier, both mentally and physically. We provide services that chauffeur older adults to meet their mobility needs. … Our services support the needs of older adults” (participant 4, private transport operator).

## 5. Discussion

The operation of public chauffeuring and companionship for older adults is described in Figure 1. The basis for providing such services depends on four main elements: availability, accessibility, safety, and affordability. At present, there are already many transport providers and local authorities who understand the need to provide on-demand driver services, such as private chauffeuring, for older adults. This is evident from the available policies and vehicles ready to accommodate older adults. Some of these transport providers include companionship as part of their chargeable services. Others do not have such provisions but may offer companionship services to older adults out of courtesy. Some specific companionship services require professional chauffeurs or drivers to attend to older adults. As iterated by one transport operator, “Sometimes, it takes time to confirm a request from our passengers. Some specific services require freelancers (drivers) who have experience and knowledge in handling older adults” (participant 5, private transport operators). Although the transport operators did not mention that passengers were denied services due to the unavailability of chauffeurs or drivers to help with older adults’ special needs, it could nevertheless be suggested that more professional chauffeurs or drivers who are experienced and qualified to handle older adults be trained and recruited to expand such services.

In addition to the availability of services, the accessibility of such services is also important to ensure that private chauffeuring and companionship services can be extended to older adults. Here, accessibility is measured as the ease of reservation for services when needed. Understanding that many older adults are not proficient at using smartphones or phone applications for making reservations, some private operators accept reservations through phone calls. This was confirmed in earlier studies [39,71], which found that physical difficulties and lack of comfort and familiarity with technology could be the reason older adults are disconnected from new transport technology. Reservations should, thus, not be restricted to only apps, even though the world is advancing in technology and digitalization. On-site queuing systems should be maintained, as well as more traditional in-advance reservation methods through platforms such as call centers, online reservations, telephone calls, and messages via SMS or WhatsApp. Services are limited to only transportation, and do not include companionship based on a first-come-first-serve basis, or depend on the availability of vehicles at the on-site queuing system. More specific services that specify a time of pick-up, destination, or multiple destinations, as well as specific services for companionship to assist older persons with their travel needs, need to be agreed upon during reservation.

**Figure 1 ijerph-20-02720-f001:**
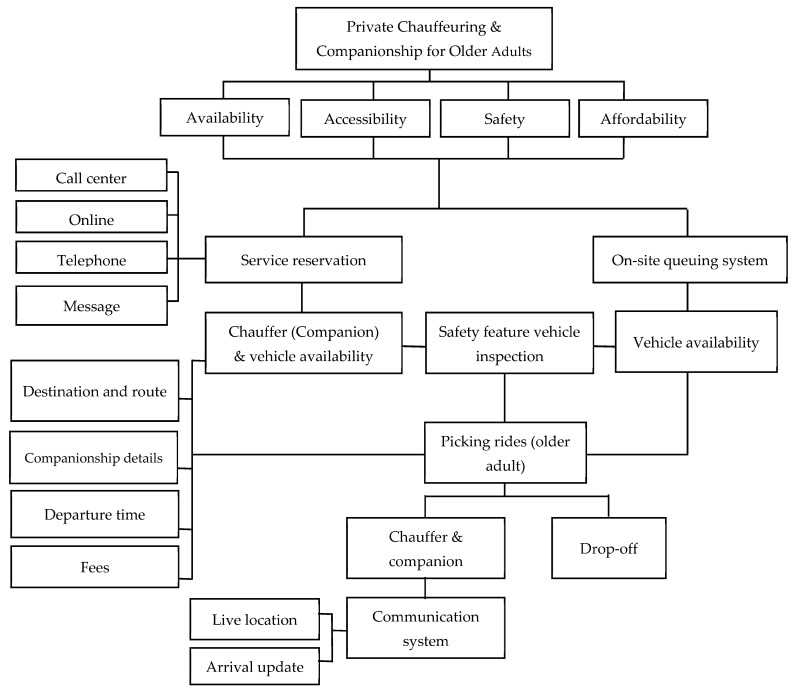
Private chauffeuring and companionship services for older adults. Note. Source: Authors, adapted from Wang and Shi [72].

Safety is one of the most important features of private chauffeuring and companionship services. Although many transport operators note that there are no special safety features for older adults, vehicle owners and drivers ensure that the vehicle is fit and passes the safety features required by law for all their passengers. Private services also include additional safety measures, such as live location sharing with family members to ensure that the safety of the older adults is well taken care of. Under any circumstances, there should be open communication between drivers and family members to update the whereabouts of the older adults.

As distance and requirements vary based on personal needs, travel costs are also agreed upon during reservation. As this is an on-demand service, the price can be expensive and unaffordable for older adults. It was mentioned in the interviews that older adults unable to afford the quote prices would be referred to other transport operators, which could lead to denying their ability and rights regarding independent travel. As travel is necessary for many social interactions that are important for wellbeing, as emphasized by [27], a concession is necessary to reduce transportation costs so that older adults can engage in social activities and maintain healthy social relationships.

The aspects of mobility catered to by private chauffeuring and companionship services can enhance older adults’ wellbeing through many aspects, including inclusivity, empowerment, mental health, physical health, and social engagement (Figure 2). Inclusivity means taking into consideration the needs and requirements of older adults in government transportation policies and developmental plans. Malaysia’s national transport policy (2019–2030) emphasizes the concept of inclusivity to provide consumer-friendly services to various groups, such as the disabled, older persons, women, children, and consumers in rural areas. The inclusion of inclusivity in the policy aims to overcome a lack of facilities and services for older adults. Inclusivity also means that older adults continue to actively participate in daily activities at the community level. Active participation reduces one’s sense of isolation and increases older adults’ involvement in society. As defined by [2], a key determinant of active/healthy ageing and wellbeing is active participation and involvement in daily activities in the community. As older adults can arrange a specific pick-up and drop-off time with the transport operators, they have more control over their travel decisions. This enhances their autonomy and independence.

Some older adults make specific personal arrangements with their chauffeur or drivers for additional services. One transport operator explained that “Sometimes older adults ask me for a stopover to do groceries or buy lunch. Sometimes I am the one who runs the errands for them. At other times, I only wait for them in the car” (participant 3, private transport operators). Older adults are empowered to make decisions, travel on their own time and for their own needs, and plan for their activities without interference. Undeniably, active participation means that older adults can retain and improve their mental and physical health. As emphasized by [73], mobility is linked to better physical and mental health. Older adults who are actively mobile are less lonely and more connected to their community, leading to improved wellbeing.

**Figure 2 ijerph-20-02720-f002:**
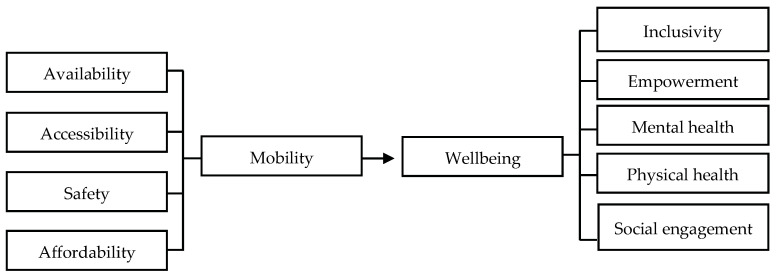
The relationship between private chauffeuring and companionship with the wellbeing of older adults. Note. Source: Authors, adapted from Cuignet, et al. [74].

## 6. Conclusions

An elderly-friendly transportation system is essential to allow older adults to be mobile, travel from one place to another to meet their basic needs and health care, continue actively participating in society, maintain social networks, and retain autonomy and independence in life. Mobility difficulties faced by older adults due to public transportation’s limited routes and services and driving cessation call for a need for more old-age-friendly transportation services. Private chauffeuring and companionship are alternative modes of transport that allow older travelers to travel based on their needs with a companion to cater to their safety and specific requirements. Effective interventions for sustainable and efficient private chauffeuring and companionship should include improving the availability, accessibility, affordability, and safety of such services to enhance older adults’ wellbeing.

The findings of this study are limited to the perception of transport operators and government agencies who provide private chauffeuring and companionship services to older adults. While their views on the operations of such services could be considered comprehensive in explaining the supply side of the services, their perceptions of wellbeing are limited to their experience in handling older adult passengers. Hence, there is a need to extend the study to obtain the older adults’ own perception of such services for better policy formulation.

## Figures and Tables

**Table 1 ijerph-20-02720-t001:** Discussion topics and their respective key questions.

Discussion Topic	Key Questions
Views on private chauffeuring and companionship to fulfil older persons’ mobility needs,	Topic 1: Availability of services What is the type of services provided to older adults? Do the services include private chauffeuring? What about companion services? What additional services are provided to older adults? What motivates the provision of such services? Is provision of such services take into account the issues of urbanisation and ageing society? Topic 2: Accessibility of services How are the services offered extended to the older adults? How accessible are such services to older adults? Topic 3: Safety What are the specific safety features of the transport services for older adults? Topic 4: Affordability What steps have been taken to ensure that transport services are affordable for older adults? Do you provide concession? How much? How does this affect services demand and operational costs
Challenges faced in meeting older persons’ mobility needs	What are the challenges faced in fulfilling the demand and needs of older adults? How does the government help in mitigating the identified challenges?

## Data Availability

Not applicable.

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
