# Peer review of "Transport, Mobility and the Wellbeing of Older Adults: An Exploration of Private Chauffeuring and Companionship Services in Malaysia"

_ijerph, 2023, doi:10.3390/ijerph20032720_

Round 1

Reviewer 1 Report

The article analyzes the perception of transport operators, government agencies, and city councils that offer private chauffeuring and companionship services for older adults and summarizes their views regarding availability, accessibility, and affordability for older adults' mobility needs. The reviewer finds the article interesting but considers this a qualitative analysis lacking a supporting methodology and scientific evidence to support the results.

Reviewer's comments:

Lines 10-11: First sentence's meaning is not clear. It is too vague and confusing. Please consider rephrasing.

Lines 20-22: Please include in the abstract a very short description of your results and a short reference to their value for the scientific community (as connected to the topic of the journal).

3) Regarding the introduction: Very well-written introduction, but please include a reference to the following sections (what each section includes). You are suggested to include a reference to the methodology and results (and how they connect with what you described in the introduction).

4) Regarding your methodology section: Please include a literature review (either as a separate section or as a sub-section of the methodology) that connects the methodology with previous work in the literature. This is a very central piece of work that will enable readers to understand the bigger picture, understand the scientific evidence, and be able to reproduce results.

5) Regarding the Methodology and Findings sections, please consider answering these questions in your manuscript: What is the scientific method used for deriving the findings? What is the methodology behind the scientific method? Where has the scientific method been previously used? How is it re-used or extended in this paper? What is the scientific value of the Findings? What are other scientific findings connected to your results in this report?

Author Response

Thank you for your kind and constructive comments. We have done our best to address all comments as shown in the manuscript and detailed out in the Response to Reviewer 1 file. 

Reviewer 2 Report

Transport, Mobility and the Wellbeing of Older Adults: An Exploration of Private Chauffeuring and Companionship Services.

Comments and Suggestions:
1. Title: The title must include the word Malaysia as the location of this study/ research.

Transport, Mobility and the Wellbeing of Older Adults: An Exploration of Private
Chauffeuring and Companionship Services in Malaysia

2. This topic is original and relevant in this field. This study is very important because
it shows the needs of the elderly not only in Malaysia but all over the world. Malaysia
will face an aging country in 2030.

3. The research methodology has been explained very well and is easy to understand.

4. All references are appropriate and up to date.

This article can be accepted for publication. All discussions have been addressed well and easy to understand. All the references used are also relevant and up-to-date in accordance with the entire content of the discussion.

Author Response

Thank you for your kind and constructive comments. We have done our best to address all comments as shown in the manuscript and detailed out in the Response to Reviewer 2 file. 

Reviewer 3 Report

Paper title: Transport, Mobility and the Wellbeing of Older Adults: An Exploration of Private Chauffeuring and Companionship Services

Journal: Int. J. Environ. Res. Public Health

Summary: In this empirical paper, the authors present the results of a qualitative approach based on focus groups with different stakeholders connected with improving the mobility of the elderly population. Their aim is to understand characteristics of the private chauffeuring and companionship services for the elderly and infer the impact of the service of the elderly’s well-being. After presenting a brief but well-rounded literature review on the role of transport to the well-being of older adults, the authors succinctly introduce their methodology and present results indicating that private chauffeuring and companionship service effective at improving the elderly’s well-being needs to address its accessibility, affordability, availability, and safety.

Broad Comments The paper is well written, well-structured, and quite interesting. The authors succinctly present their methodology but never express their main research question in the paper, leaving the reader to guess what problem is it that they seek to investigate. The methodology section which appears incomplete or misleading prevents the reviewer from judging if the results are scientifically fit or not. A more complete methodology would be required for us to affirmatively do so. The authors present their findings in a well-organized manner all the while highlighting the qualitative data extracted through their work by quoting the participants directly. We find the conclusions very interesting and showcasing an interesting program. The conclusions appear sound and stemming from the analysis but could not be confirmed with the current insufficiently detailed methodology. The authors clearly address the limits of their approach which did not include elderly users of the system as participants in the research.

Specific Comments

Major issues

1. The methodology section is incomplete. The authors briefly discuss their set-up for the focus group, but do not specify how they recruited the participants nor what was the protocol used during the focus groups. I consider it important to briefly introduce the different questions on the Google slides or add them as an appendix to the paper. The methodology fails to indicate how the analysis is performed. It is missing quite a few details on the analysis and interpretation of the focus group material. The authors only specify that they “a transcription of the session was uploaded to NVivo 12 for analysis”, what kind of coding scheme, inducted or deducted codes, how did they ensure the inter-coder coherence or was all the coding done by only one of the researchers? All these questions should be answered in a complete methodology section.

2. The methodology section appears inaccurate or misleading and is unclear. On line 134 the authors specify that they “adopt the measures of wellbeing following 42” (42 being Ingersoll-Dayton et al., 2004). However, Ingersoll-Dayton et al. relied exclusively on elderly participants (not transportation stakeholders) to measure the elderly’s well-being and using a mixed approach which included qualitative as well as quantitative approach. The allusion to this source is confusing for the peer reviewer and appears misleading as the methodology used by the authors does not involve any elderly participants nor any quantitative testing of well-being. This appears misleading and should be explained in the paper by the authors. How the approach of Ingersoll-Dayton et al., 2004 was adopted in their research. The next sentence on lines 134 to 137 also needs to be clarified, why were this proxy used and how does it make sense with the author’s proposed methodology?

Minor Issues

3. No results are presented in the abstract.

4. On line 66 the authors refer to 29 (Schwanen et al., 2015) in a passage discussing social exclusion of older adults making a very broad link between the two. This seems particular as Schwanen et al.’s research is broad and not targeted at the elderly and their conclusions are rather generalists. They discuss the elderly situation only in their literature review and not as part of their findings or conclusions of their paper. The authors might actually be referring to Ryser and Halseth’s (2012) who are cited by Schwanen et al., 2015 as it related to a specific case in Canada regarding the elderly.

5. On line 407 the authors state that “active participation would mean that older adults can retain and improve their mental and physical health”. The next sentence on line 408 refers to 46 (Musselwhite et al. 2021) to seemingly to support their claim. However, the article by Musselwhite et al. 2021 is a broad article about the general populations and make no claims specific to the older population. This should be specified by the authors in their sentence on line 408 or they should find a reference specifically looking at the older population.

Author Response

Thank you for your kind and constructive comments. We have done our best to address all comments as shown in the manuscript and detailed out in the Response to Reviewer 3 file. 

Reviewer 4 Report

The article presents an innovative, relevant and current theme and is well structured in terms of the proposed methodology. The introduction is well prepared, it makes clear the relevance of the study and the knowledge gap that it wants to clarify. The objective is explained clearly and pertinently.

The methodology was well structured and meets the scientific criteria in a rigorous and adequate manner. The results are well presented and respond to the proposed objective. The discussion presents references consistent with the study. And the conclusion was well written and responds to the purpose of the study.

Therefore, I issue a favorable opinion to the publication of the article

Author Response

Thank you for your kind and constructive comments. We have done our best to address all comments as shown in the manuscript and detailed out in the Response to Reviewer 4 file. 
